# Health workers' perspectives on asthma care coordination between primary and specialised healthcare in the COVID-19 pandemic: a protocol for a qualitative study in Ecuador and Brazil

Natalia Cristina Romero [1,2] Maria Jose Cisneros-Caceres [1] Emily Granadillo,[1] Erika Aragao,[3] Adriana Romero-Sandoval,[2,4] Carolina Barbosa,[3] Ana Luiza Barreto de Oliveira,[3] Alejandro Rodriguez,[1] Gabriela Pimentel Pinheiro,[5] Alvaro Cruz,[3] Philip Cooper [1,6] Maria Rejane Ferreira da Silva[2,7]

**Correspondence to**
Dr Maria Jose Cisneros-Caceres; majocc93@gmail.com

## ABSTRACT

**Introduction** Asthma is a common long-term disorder and strategies to improve asthma control are still a challenge. Integrated delivery of health systems is critical for effective asthma care: there is limited information on experiences of care coordination for asthma from Latin America, especially on perspectives of health personnel and in the context of the COVID-19 pandemic.

**Methods and analysis** This protocol details a qualitative approach to analyse health workers' perspectives of healthcare coordination for asthma control during COVID-19 pandemic in Ecuador and Brazil, at primary and specialised levels, through in-depth semistructured interviews using a video communications platform. The analysis will identify knowledge and perspectives based on coordination of clinical information, clinical management and administrative coordination. Theoretical sampling will be used to obtain approximately equal numbers of women and men within each level of healthcare; data saturation will be used to determine sample size. Transcripts will be analysed using content-coding procedures to mark quotations related to major topics and subthemes included in the interview guide, and narrative analysis will be based on a theoretical framework for healthcare coordination to identify new themes and subthemes.

**Ethics and dissemination** Ethical approval was obtained from the ethics committees of Hospital General Docente Calderón, Quito, Ecuador; and Universidade Federal da Bahia, Salvador, Brazil. The findings of this study will be disseminated through peer-reviewed articles, conference presentations and condensed summaries for key stakeholders and partners.

## INTRODUCTION

Asthma has emerged as a major challenge for healthcare systems around the world accounting for millions of doctor visits, hundreds of thousands of emergency department visits and hospitalisations, and thousands of deaths.[1 2] Among those who have

> **Strengths and limitations of this study**
>
> ► This qualitative study protocol is the first to focus on the perspectives of healthcare workers on asthma care coordination between primary and specialised levels in the context of the COVID-19 pandemic.
> ► This study will use in-depth semistructured interviews of healthcare workers from four cities in Ecuador and Brazil to compare perspectives on asthma care and healthcare system coordination between locations.
> ► Qualitative interviews will be done over the period of a year during the COVID-19 pandemic that likely will affect coordination of asthma care.
> ► Although our recruitment method should identify key informants, we could not involve the best representatives from primary and specialised care levels.

asthma, children are the most affected group and asthma is now the most common chronic disease of childhood.[3] While our knowledge on asthma has increased, we still have an incomplete understanding of its causes: factors including environmental exposures, changes in lifestyles and host genetics are likely to be important.[1]

There are wide variations in the prevalence of asthma symptoms between countries.[4] However, trends in asthma mortality and hospitalisation have shown a progressive reduction or stabilisation, especially in high-income countries (HICs), where asthma prevalence reached epidemic levels four decades ago.[5] Presently, although the prevalence of asthma has stabilised in HICs, it appears to be increasing in some low/middle-income countries (LMICs), regions that account for more than 80% of asthma deaths worldwide.

Nonetheless, many cases of severe asthma and asthma deaths are preventable through optimal management using medications to relieve and control the disease and improvements in healthcare coordination (HCC).[1 6]

HCC has been shown to be a promising strategy to improve asthma management and control in HICs: improved care coordination results in reduced asthma symptoms and urgent healthcare utilisation which is cost-effective.[7] However, in LMICs, HCC remains a challenge for patients with asthma because most of the health systems in these regions are characterised by fragmentary organisation and a lack of resources.[8] Additionally, medical therapy is often hindered by fragmented disease-specific approaches to management (fostered by single-disease guidelines), use of acute care to manage chronic diseases and inadequate integration of care across multiple levels, especially from hospital to home.[9 10]

In Latin America (LA), HCC has been characterised by a reconstruction of the self-identity concept.[11] There are few reports of HCC implementation,[12 13] while asthma management and control are uncoordinated in most health systems, showing high rates of acute exacerbation and use of urgent care with high hospitalisation rates.[14 15]

The outbreak of infections with the novel coronavirus, SARS-CoV-2, in Central China, in December 2019 marked the beginning of the COVID-19 pandemic.[16] Measures to reduce viral transmission were adopted in Ecuador and Brazil from March 2020 with enforced social isolation and restricted movement at local or national levels interfering with opportunities for interactions with healthcare. Although lockdowns had been lifted by the second half of 2020, there was still limited access to emergency rooms and outpatient care at primary and specialised levels for non-COVID-19 consults,[17] a situation that continued into the first quarter of 2021.

This protocol details the methodological and analytical considerations using a qualitative approach in a study designed to understand and characterise healthcare workers' perspectives of care coordination for asthma control between primary and specialised levels of public healthcare systems in Ecuador and Brazil during the COVID-19 pandemic.

Based on the characteristics of the qualitative research, we are flexible with the completion date for the study. Still holding in-depth interviews as well as data collecting until we reach theoretical saturation, expecting to have results by the first quarter of 2022.

## METHODOLOGY
### Study context
The present study is part of a study ('Asthma Attacks Causes and Prevention Study in Urban Latin America (ATTACK)') developed in 2019 in collaboration with partners from the UK, Brazil and Ecuador. The study seeks to understand better the causality of asthma attacks and of their recurrence and optimise strategies to improve asthma control and prevent asthma attacks in LA.

### Study design
A qualitative research design will be used to explore health workers' perspective in asthma control to characterise and compare healthcare networks in Ecuador and Brazil during the COVID-19 pandemic. Data will be analysed using a narrative approach focusing on the participants' perception, whereby the researcher and the participant develop jointly results from an interactive conversation.[18 19] Narrative theory aims to understand the succession of facts, situations, phenomena, processes, and events where thoughts, feelings, emotions, and interactions are involved, through the experiences told by those who experienced them.[20]

### Theoretical framework
The present study is based on the theoretical framework of Integrated Health Care Networks (IHCN).[21] Health systems must be understood as IHCN, defined as networks of organisations that provide or make arrangements to provide equitable and integrated health services to defined populations that are accountable for their clinical and economic impacts as well as the health of the populations they serve.[21] Among the IHCN objectives, we can distinguish two groups: (1) intermediate objectives: (a) access, (b) care coordination, (c) continuous care; (2) final objectives: (a) equity in access and (b) efficiency.

In the present study, we focus our efforts on the analysis of types of care coordination, represented by clinical information coordination, coordination of clinical management and administrative coordination.[22]

### Clinical information coordination
This focuses on referrals between care levels and use of information from previous episodes and biopsychosocial situations relevant to the current patient consultation. The way in which information is transmitted between healthcare professionals affects how current events are linked to previous events and how the current consult is thus adapted to the needs of the patient. For this study, this will be divided into two dimensions: clinical and psychosocial information referral where attributes consisted of documents for the referral, agile and timely access to information, pertinent content of the information and registry of the information by professionals, and the use of the information whose attributes are: transfer of information from the consultation and the incorporation of information into the clinical practice.[22]

### Coordination of clinical management
The provision of healthcare in a sequential and complementary way, with a clinical management plan shared by the different care levels and participating services. It is defined by three dimensions: care coherence (similar approximations among the professionals from different levels), accessibility among levels (provision without interruptions) and accurate follow-up of the patient among levels (in the transitions from one care level to another).[22] The first dimension being the follow-up of

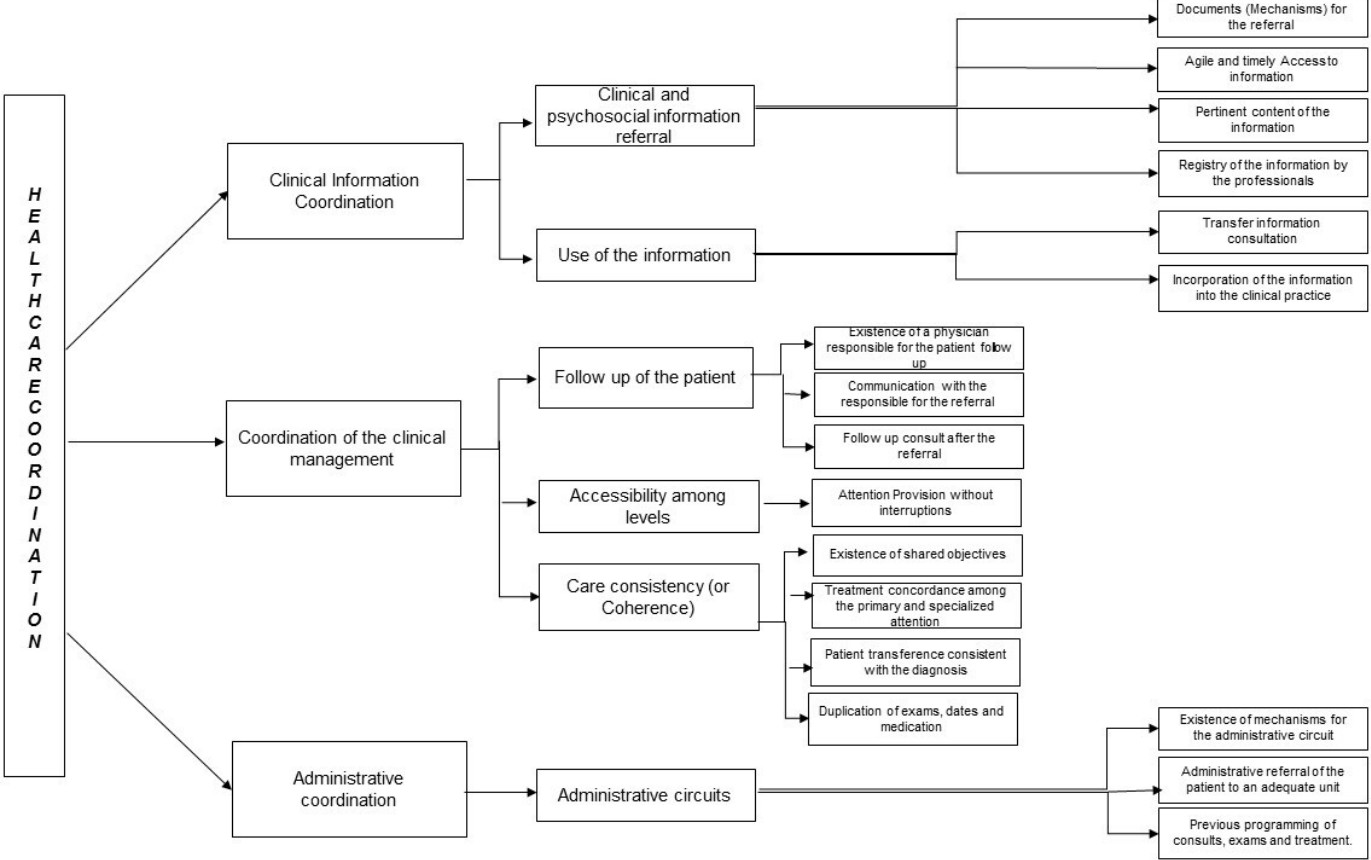

**Figure 1** Theoretical framework for healthcare coordination. The theoretical framework is shown by type of coordination and attributes and dimensions of each.

the patient that includes attributes such as existence of a physician responsible for patient follow-up, communication with those responsible for making the referral and having a follow-up consult after the referral. The second dimension, accessibility among levels, focuses on the provision of attention without interruption; and the third dimension, care consistency (or coherence), that centres on attributes like: shared objectives, treatment concordance between primary and specialised labels, patient transference consistent with the diagnosis and the duplication of examinations, dates, and medication.

### Administrative coordination

This is defined as the coordination of patients' access along the care continuum according to needs. It works with the administrative circuits dimension, where attributes such as existence of mechanisms for the administrative circuit, administrative referral of the patient to a suitable unit and previous programming of consults, examinations and treatment, are studied. The importance of HCC among health system levels constitutes a strategy to achieve continuous care, reduce costs and improve quality of care. The perception of the information dimensions and management of the HCC relies on factors of the professionals, existence and use of coordination mechanisms and organisational factors.[22] The theoretical framework is shown in figure 1.

### Study area

The study will be conducted in three cities in Ecuador and one city in Brazil. Ecuador is an upper middle-income country with a per capita income of $6110. According to the WHO's Global Health Workforce Statistics, the physicians per 1000 people for Ecuador and Brazil are around 2.2, and the nurses 10.1 for Brazil and 2.5 for Ecuador.[23 24] In 2009, the Integral Public Health Network became a constitutional mandate, aiming to develop collaborations between private and public sectors. The public institutions offer healthcare services to the entire population and are divided into different levels of care and distributed geographically.[25] Social security health institutions offer services only to the affiliated salaried population. The private sector includes for-profit entities (hospitals, clinics, dispensaries, doctor's offices, pharmacies and prepaid health insurance companies) which are generally located in the main cities of the country. Quito, the capital of Ecuador and one of the cities where the study will be conducted, is located in the Andean region at an altitude of 2800 m, is one of the most populous Ecuadorian urban areas with 2 781 641 inhabitants, and has greater socioeconomic and health indicators than the national average. The two other cities in the study are: Cuenca, located at altitude of 2550 m with a population of 329 928 inhabitants, and Portoviejo located at altitude of

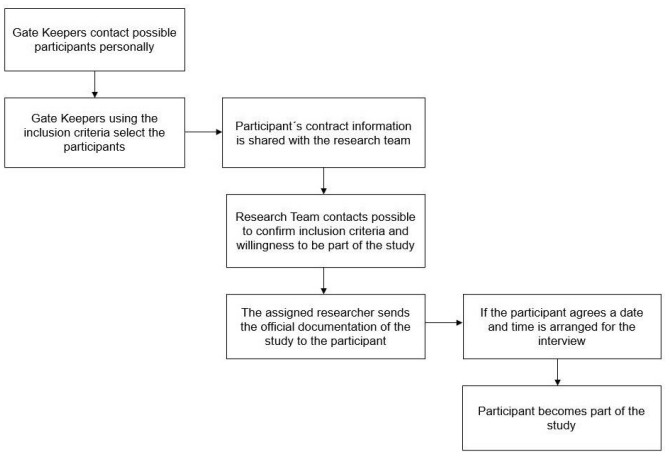

**Figure 2** Flow diagram for recruitment showing how health actors become participants.

53 m with a population of 206 682 inhabitants.[26–28] Public healthcare in Ecuadorian cities is divided geographically into sanitary districts: Quito with nine, Cuenca with two, and Portoviejo with one. Districts chosen for study were 17D02, 01D01, and 13D01 in Quito, Cuenca, and Portoviejo, respectively.

Brazil is an upper middle-income country with a per capita income of $9130.[26] The Brazilian health system has been characterised by the principle of universal healthcare access mandated by its constitution since 1988. The Unique Health System of Brazil (SUS) has different levels of care including primary and specialised in which the public and private sector participate and is organised hierarchically and geographically with the aim of supporting SUS to the benefit of the population.[29] The municipality of Salvador, capital of the state of Bahia, is the most populous in Northeast Brazil with a population of 2 953 986 inhabitants living in the urban area.[30] The organisation of public healthcare in Salvador is territorial and divided into 12 sanitary districts. In Salvador, the railway district was chosen.

Both in Ecuador and Brazil, the healthcare models include general principles of distribution of tasks between the community level, first level and the specialist level, through referral and counter-referral, minimising duplication of functions and competition between levels. Although guidelines are provided by each national health ministry, we would like to understand better how this coordination between levels functions using a chronic disease such as asthma as a model condition.

### Participants: study subjects

Health actors are defined as individuals or groups with an interest in the health system.[31] Family doctors, nurses, specialists, and public health representatives working in primary, secondary, and tertiary care levels involved in the care of patients with asthma will be invited to participate. Gatekeepers which are health professionals who work on a regular basis with patients with asthma and within the healthcare system in both countries will be responsible for the first contact. We will recruit a minimum of 20 participants per country to achieve speech saturation. Inclusion criteria are (1) Ecuadorian and Brazilian health professionals with at least 6 months of experience; (2) public health representatives with a wide range of experience; (3) managers with senior positions (ie, directors) at levels of healthcare. A flow diagram for participant recruitment is shown in figure 2.

### Patient and public involvement

No patient involved.

### Data collection, instruments and procedures

#### Data collection

In-depth interviews will be used for data collection to explore health workers' experiences and views on HCC for asthma. Contacts due to the COVID-19 pandemic will be via cell phone, WhatsApp or email. Study documentation shared with the participants includes an information letter, informed consent for recording of video calls and for participation in the study, and the participant information sheet.

Participants will be asked to openly describe their experiences and vision about care coordination. The interview will be conducted by two persons from the study team composed of an interviewer and an observer. Interviewers will use a semistructured guideline including the following topics:

1. Asthma management in primary and specialised care levels.
2. Availability and use of asthma guidelines or care protocols by health workers.
3. Care coordination for patients with asthma between primary and specialised care levels.
4. Transfer protocols between primary, secondary and tertiary care levels.
5. Understanding of HCC and Integrated Health Network.
6. Strategies, interventions and instruments in HCC.
7. Changes and adaptations of asthma control during the COVID-19 context.

#### Procedure

Procedures are planned as follows: (a) informants will be contacted by gatekeepers in each city; (b) an introductory letter (and informed consent forms) will be provided to each informant explaining the study; (c) if the participant agrees to be part of the study and signs consent, the following will be done; (d) each interview, researcher and participant will be assigned an alphanumeric unique identifier code to ensure confidentiality; (e) each interview will last approximately 50 min; and (f) interviews will be audio-recorded and transcribed.

#### Data analysis tools and procedures

This study is based on narrative analysis of manifest content through a priori categories and using constant comparisons between the speech of each of the defined profiles as well as emerging categories. Narrative analysis

will be segmented by case, type of actor and topic. The content of each interview will be read by the researcher, identifying significant fragments, classified by type of informant, with the purpose of identifying similarities and differences. We will use descriptive data regarding the health systems of Brazil and Ecuador to establish comparisons between the countries (comparative table). In this study, we will produce triangulations among informants, researchers and the theoretical framework. Analysis will use NVivo (Neu, QSR International).

The triangulation process will allow us to work with three different levels of analysis. At the first level, the significant phrases will be codified using the theoretical framework from the participants' transcript interviews, aiming to saturate speech. At the second level, two different researchers from each country will analyse the chosen significant phrases and decide if the significant phrases correspond to the categories or not. After this process, the second level and first level researchers meet up in a discussion of the results before moving on to the third level. At the third level with the consolidated information from the first two levels, the experts make an analysis of the significant phrases using the theoretical framework and comments made at other levels before deciding if a category is saturated. The information is once again discussed between the three levels before accepting it as a significant phrase.

Extensive training was provided to the researchers participating in the study at all levels aiming to standardise concepts and procedures.

For the construction of the analysis and reporting of results, we will follow the recommendations Standards for Reporting Qualitative Research guideline.[32]

## ETHICS AND DISSEMINATION

The protocol was approved by the Ethics Committees of the Hospital General Docente Calderón (CEISH-HGDC 2019-001) in Ecuador and Faculdade de Medicina da Bahia da Universidade Federal da Bahia (CAAE: 04057518.0.0000.5577) in Brazil. The study will be done in accordance with the guidelines of the World Medical Association and the Declaration of Helsinki.

This study forms a part of the National Institute of Health Research Global Health Research Group 'ATTACK'. We acknowledge that all participants must be available and be capable of participating in the study and that there is a small chance that the interviewees could create a role of power among the interviewers if the purpose of the research is not understood. To tackle these issues, we will inform all participants about the purpose of the research and will obtain written informed consent from each participant. Besides designing an interview guide, we have had to consider how to formulate questions in a way that minimises feelings of discomfort, judgement and criticism among participants.

Our findings will be shared first with the participants of the study and researchers in collaborating institutions.

The result will be presented as practical recommendations in each country (Ecuador and Brazil) at policy briefings and forums of healthcare professionals. Our results will also be presented at seminars, academic conferences and in peer-reviewed scientific articles.

**Author affiliations**
¹School of Medicine, International University of Ecuador, Quito, Ecuador
²Department of Medicine, Red Groups in Latin America and Africa, Barcelona, Spain
³Collective Health Institute, Federal University of Bahia, Salvador, Brazil
⁴School of Basic Sciences, International University of Ecuador, Quito, Ecuador
⁵Department of Medicine, ProAR Foundation, Salvador, Brazil
⁶Institute of Infection and Immunity, St George's University of London, London, UK
⁷School of Medicine, UPE, Recife, Brazil

**Contributors** All have been an active part of the analysis and interpretation. All authors have contributed to the drafting of the work and revising important content for its elaboration. All have approved the final version to be published, and agree to be accountable for all aspects of the work.

**Funding** This study forms part of the Asthma ATTACK Study and was funded through the National Institute of Health Research (NIHR) Global Health Research Group at St George's University of London by the NIHR, UK (grant 17/63/62) using Official Development Assistance (ODA) funding.

**Competing interests** None declared.

**Patient and public involvement** Patients and/or the public were not involved in the design, or conduct, or reporting, or dissemination plans of this research.

**Patient consent for publication** Not required.

**Provenance and peer review** Not commissioned; externally peer reviewed.

**ORCID iDs**
Natalia Cristina Romero http://orcid.org/0000-0001-6881-6581
Maria Jose Cisneros-Caceres http://orcid.org/0000-0001-5996-5147
Philip Cooper http://orcid.org/0000-0002-6770-6871

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
