## [Reviewer comments · BMJ Open]

ARTICLE DETAILS

TITLE (PROVISIONAL)	Health worker's perspectives on asthma care coordination between primary and specialized health care in the COVID-19 pandemic: a protocol for a qualitative study in Ecuador and Brazil.
AUTHORS	Romero, Natalia; Cisneros-Caceres, Maria; Granadillo, Emily; Aragao, Erika; Romero-Sandoval, Adriana; Barbosa, Carolina; Barreto de Oliveira, Ana Luiza; Rodriguez, Alejandro; Pinheiro, Gabriela; Cruz, Alvaro; Cooper, Philip; Ferreira da Silva, Maria Rejane

VERSION 1 – REVIEW

REVIEWER	Mroueh, Salman American University of Beirut, Pediatrics and Adolescent Medicine
REVIEW RETURNED	19-Jul-2021

GENERAL COMMENTS	This is a straightforward study to answer a research question. The protocol is well written and the methods well described. The study is realistically achievable within the proposed timeframe.
--

REVIEWER	Soyiri, Ireneous University of Hull, Hull York Medical School
REVIEW RETURNED	20-Aug-2021

GENERAL COMMENTS	The proposed study on asthma care coordination between primary and specialized health care in the COVID-19 pandemic in Ecuador and Brazil, is well written and succinctly presented. The design is well thought-through. I wish the study team all the best as the pursue this area of research, which would hopefully yield results that may contribute a number of actionable local/national policy recommendations.
--

REVIEWER	Jaakkimainen, R. Liisa Institute for Clinical Evaluative Sciences
REVIEW RETURNED	30-Aug-2021

GENERAL COMMENTS	This is a proposal for a qualitative study examining the perspective of healthcare workers in Brazil and Ecuador on the coordination of asthma care and the impact of the COVID-19 pandemic on the coordination of asthma care. This is an important topic as an understanding of chronic disease coordination could extrapolated to many conditions outside of asthma and this information could be used in several healthcare systems. There is a lot of data that the research groups aims to extract and a qualitative method is likely the best way to get a lot of information on a variety of topics (coordination, pandemic, asthma guidelines, private care etc).
---

	The methods are well described including the context, the narrative approach, theoretical framework, different coordination foci (clinical information, clinical management and administrative) and analysis plan. Study population “area” for Ecuador and Brazil are well described, but maybe more detail on the primary care to specialist structure would be helpful. The background mentions the transfer from hospital to home, the relationship with private versus public healthcare. But how many family physicians, nurses, specialist physicians exist in each region? What proportion work in public versus private settings? I agree a minimum of 20 interviews would be needed in each country. How will they be recruited? Will the discussions include people of all ages having asthma? Are children included in the discussions? Who are the gatekeepers who will be contacting participants? How are the participants identified? Define “health actor” for a general audience. While the aim is to examine changes made as a result of the COVID19 pandemic, I didn’t get a sense of how well the health care coordination or integrated network is being used. Was it fairly-well established prior to the pandemic? Page 8 first paragraph, “1st quarter” should be written out to “first”.
--	---

VERSION 1 – AUTHOR RESPONSE

Reviewer: 1

Dr. Salman Mroueh, American University of Beirut

Comments to the Author:

This is a straightforward study to answer a research question. The protocol is well written and the methods well described. The study is realistically achievable within the proposed timeframe.

Response 2: Thanks for your comments which are deeply appreciated. We hope this study can be useful for many health personnel as well as encourage more research on this field.

Reviewer: 2

Dr. Ireneous Soyiri, University of Hull

Comments to the Author:

The proposed study on asthma care coordination between primary and specialized health care in the COVID-19 pandemic in Ecuador and Brazil, is well written and succinctly presented. The design is well thought-through. I wish the study team all the best as the pursue this area of research, which would hopefully yield results that may contribute a number of actionable local/national policy recommendations.

Response 3: Thanks for your kind words and taking the time to review our study protocol. We hope our results can contribute to knowledge in this area of research and encourage more researchers to pursue this research area. We hope this can be the start of policy recommendation to improve health and highlight the relevance of healthcare coordination to provide best possible care for patients.

Reviewer: 3

Dr. R. Liisa Jaakkimainen, Institute for Clinical Evaluative Sciences

This is a proposal for a qualitative study examining the perspective of healthcare workers in Brazil and Ecuador on the coordination of asthma care and the impact of the COVID-19 pandemic on the coordination of asthma care. This is an important topic as an understanding of chronic disease

coordination could be extrapolated to many conditions outside of asthma and this information could be used in several healthcare systems.¹

There is a lot of data that the research groups aim to extract and a qualitative method is likely the best way to get a lot of information on a variety of topics (coordination, pandemic, asthma guidelines, private care etc).

The methods are well described including the context, the narrative approach, theoretical framework, different coordination foci (clinical information, clinical management and administrative) and analysis plan.

Response 4: Thank you for your comments. We have added the following sentence to page 10: "We will follow SRQR (Standards for Reporting Qualitative Research) recommendations for the construction of the analysis and reporting of results (O'Brien et al, 2014)."

- Study population "area" for Ecuador and Brazil are well described, but maybe more detail on the primary care to specialist structure would be helpful.
- The background mentions the transfer from hospital to home, the relationship with private versus public healthcare. But how many family physicians, nurses, specialist physicians exist in each region?
- What proportion work in public versus private settings?

Response 5: Thanks for your comments dear Dr. R. Liisa, we appreciate your suggestions as well as questions.

Regarding your question about the primary care to specialist structure, both in Ecuador and Brazil, the health care models include general principles of distribution of tasks between the community level, first level and the specialist level, through referral and counter-referral, minimizing duplication of functions and competition between levels. Although guidelines are provided by each national health ministry, we would like to understand better how this coordination between levels functions using a chronic disease such as asthma as a model condition. This paragraph has been included in the reviewed version.

In addition, we have included this phrase: The number of physicians per 1000 people for Ecuador and Brazil is 2.2, while number of nurses is 10.1 for Brazil and 2.5 for Ecuador (data according to the World Health Organization's Global Health Workforce Statistics).

Finally, about your question of the proportion of work in public versus private settings, we are not able to provide specific number as this information is not available through the national public register. In Ecuador, the administrative information for health professionals working in public and private health institutions does not differentiate between public and private.

- I agree a minimum of 20 interviews would be needed in each country. How will they be recruited?

Response 6: This information is provided in the Procedures section of the text on page 9 as follows: "Procedures are planned as follows: a) informants will be contacted by gatekeepers in each city; b) an introductory letter (and informed consent forms) will be provided to each informant explaining the study; c) if the participant agrees to be part in the study and signs consent, the following will be done; d) each interview, researcher and participant will be assigned an alphanumeric unique identifier code to ensure confidentiality; e) each interview will last approximately 50 minutes; and f) interviews will be audio-recorded and transcribed".

- Will the discussions include people of all ages having asthma? Are children included in the discussions?

Response 7: Information to this question can be found in the Study design section on page 5 as follows:

“A qualitative research design will be used to explore health workers’ perspective in asthma control to characterize and compare health care networks in Ecuador and Brazil during the COVID-19 pandemic. Data will be analysed using a narrative approach focusing on the participants perception, whereby the researcher and the participant develop jointly results from an interactive conversation (18,19)”.

The population for this study is neither the patients nor their caregivers. Health workers can attend adult patients or children with asthma.

Further information on participants can be found in the Participants: Study subjects’ section on page 8 as follows:

“We will recruit a minimum of 20 participants per country to achieve speech saturation. Inclusion criteria are: a) Ecuadorian and Brazilian health professionals with at least 6 months experience; b) public health representatives with a wide range of experience c) managers with senior positions (i.e directors) at levels of health care”.

- Who are the gatekeepers who will be contacting participants? How are the participants identified?

Response 8: The gatekeepers are health professionals that work on a regular basis with asthma patients and within the healthcare system in both countries. This explanation will be added to the manuscript (see page 8).

Participants are being identified by the inclusion criteria (page 8), which are:

“a) Ecuadorian and Brazilian health professionals with at least 6 months experience; b) public health representatives with a wide range of experience 3) managers with senior positions (i.e directors) at levels of health care”.

- Define “health actor” for a general audience.

Response 9: A health actor is defined as an individual or group with an interest in the health system, this has been included in the text. Thanks for your recommendation.

- While the aim is to examine changes made as a result of the COVID19 pandemic, I didn’t get a sense of how well the health care coordination or integrated network is being used. Was it fairly-well established prior to the pandemic?

Response 10: Thank you for highlighting this. Information on this is provided in the introduction (page 4):

“However, in LMICs health care coordination remains a challenge for asthma patients because most of the health systems in these regions are characterized by fragmentary organization and a lack of resources (8). Additionally, medical therapy is often hindered by fragmented disease-specific approaches to management (fostered by single-disease guidelines), use of acute care to manage chronic diseases, and inadequate integration of care across multiple levels, especially from hospital to home (9,10).

In Latin America (LA), HCC has been characterized by a reconstruction of the self-identity concept (11). There are few reports of HCC implementation (12,13), while asthma management and control are uncoordinated in most health systems, showing high rates of acute exacerbations, and use of urgent care with high hospitalization rates (14,15)”.

We also want to emphasize in the strengths and limitations section that states (page 3): “This qualitative study protocol is the first to focus on the perspectives of healthcare workers on asthma care coordination between primary and specialized levels in the context of the COVID-19 pandemic”.

Page 8 first paragraph, “1st quarter” should be written out to “first”

Response 11: Thank you for your observation. We have modified in the new version.

VERSION 2 – REVIEW

REVIEWER	Jaakkimainen, R. Liisa Institute for Clinical Evaluative Sciences
REVIEW RETURNED	06-Oct-2021
GENERAL COMMENTS	Thank you for your responses to my questions and the edits to your paper. It think you have addressed them very well.